# Synthesis, Optical Characterization in Solution and Solid-State, and DFT Calculations of 3-Acetyl and 3-(1′-(2′-Phenylhydrazono)ethyl)-coumarin-(7)-substituted Derivatives

**DOI:** 10.3390/molecules27123677

**Published:** 2022-06-08

**Authors:** Cesar A. Villa-Martínez, Nancy E. Magaña-Vergara, Mario Rodríguez, Juan P. Mojica-Sánchez, Ángel A. Ramos-Organillo, Joaquín Barroso-Flores, Itzia I. Padilla-Martínez, Francisco J. Martínez-Martínez

**Affiliations:** 1Facultad de Ciencias Químicas, Universidad de Colima, Km 9 Carretera Coquimatlán-Colima, Coquimatlán 28400, Mexico; cvmartinez@ucol.mx (C.A.V.-M.); aaramos@ucol.mx (Á.A.R.-O.); 2CONACyT, Facultad de Ciencias Químicas, Universidad de Colima, Km 9 Carretera Coquimatlán-Colima, Coquimatlán 28400, Mexico; 3Centro de Investigaciones en Óptica A. P. 1-948, León 37000, Mexico; mrodri@cio.mx; 4Tecnológico Nacional de Mexico, Instituto Tecnológico José Mario Molina Pasquel y Henríquez Campus Tamazula de Gordiano, Carretera Tamazula-Santa Rosa No. 329, Tamazula de Gordiano 49650, Mexico; juan.mojica@tamazula.tecmm.edu.mx; 5Centro Conjunto de Investigación en Química Sustentable UAEM-UNAM, Unidad San Cayetano, Carretera Toluca-Atlacomulco Km.14.5, Toluca de Lerdo 50200, Mexico; jbarroso@unam.mx; 6Instituto de Química, Universidad Nacional Autónoma de Mexico, Circuito Exterior, Ciudad Universitaria, Ciudad de Mexico 04510, Mexico; 7Laboratorio de Química Supramolecular y Nanociencias, UPIBI, Instituto Politécnico Nacional, Av. Acueducto s/n Barrio la Laguna Ticomán, Ciudad de Mexico 07340, Mexico; ipadillamar@ipn.mx

**Keywords:** coumarins, hydrazones, optical properties, HOMO-LUMO orbitals

## Abstract

Intramolecular charge transfer (ICT) effects are responsible for the photoluminescent properties of coumarins. Hence, optical properties with different applications can be obtained by ICT modulation. Herein, four 3-acetyl-2*H*-chromen-2-ones (**1a**–**d**) and their corresponding fluorescent hybrids 3- (phenylhydrazone)-chromen-2-ones (**2a**–**d**) were synthesized in 74–65% yields. The UV-Vis data were in the 295–428 nm range. The emission depends on the substituent in position C-7 bearing electron-donating groups. Compounds **1b**–**d** showed good optical properties due to the D-π-A structural arrangement. In compounds **2a**–**d**, there is a quenching effect of fluorescence in solution. However, in the solid, an increase is shown due to an aggregation-induced emission (AIE) effect given by the rotational restraints and stacking in the crystal. Computational calculations of the HOMO-LUMO orbitals indicate high absorbance and emission values of the molecules, and gap values represent the bathochromic effect and the electronic efficiency of the compounds. Compounds **1a**–**d** and **2a**–**d** are good candidates for optical applications, such as OLEDs, organic solar cells, or fluorescence markers.

## 1. Introduction

Intramolecular charge transfer (ICT), as well as linear and nonlinear optical properties [1,2,3,4,5], are associated with organic molecules having an extended electronic π-backbone. This molecular characteristic is generally modulated by the insertion of electron-donor or -acceptor groups into the electronic π-system, to favor the ICT [6]. The typical electronic architectures that can be presented, are donor-acceptor-donor (D-A-D), donor-π-acceptor (D-π-A), or donor-π-acceptor-π-donor (D-π-A-π-D), known as push-pull type compounds. Photoluminescent compounds and materials with fluorescent and phosphorescent properties play a fundamental role in many areas, such as biology, pharmacology, chemistry, biomedicine, and optoelectronics [7,8,9,10,11,12]. For example, coumarin compounds coupled to phenylhydrazine have been reported as fluorescent probes for visualizing living tissues [13]. Photoluminescent organic compounds derived from coumarins have been applied in two-photon absorption phenomena, optical data storage, energy limitation, and the manufacture of lasers [5]. Coumarins can be obtained in good yields [9] through synthetic techniques, such as Knoevenagel, Pechmann, and Wittig reactions. Therefore, the photophysical properties of coumarins can be easily modified by substituting in positions 3, 6, 7, and 8 to obtain hybrid materials [9,14,15], which exhibit high quantum yields, important changes in Stokes shifts, and good absorption and emission profiles [16]. On the other hand, fluorescent hybrid compounds derived from hydrazines and coumarins have shown synergic effects in their fluorescent properties [17,18]. These compounds have been applied as on-off sensors [19,20], photoluminescent dyes [21], anticonvulsants [22], hydrogels [23], and antimicrobials [24].

In this work, we describe the synthesis, structural characterization, and photoluminescence proprieties of small compounds derived from 3-acetyl-2*H*-chromene-2-one (**1a**–**d**) coupled with phenylhydrazine, forming the hybrid fluorescent compounds 3-(1′-(2′-phenylhydrazono) ethyl)-2H-2-chromen-2-one (**2a**–**d**). The optical characterization of **2a**–**d** was performed in solution and in the solid-state, and the results contrasted with ab initio theoretical calculations. The UV-vis spectra of coumarins **1a,c** and **2b** have been reported in solution but not in the solid-state [25,26].

## 2. Experimental

### 2.1. Materials and Methods

Salicilaldehyde, 4-methoxy-salicylaldehyde, 4-hydroxy-salicylaldehyde, 4-diethylamino-salicylaldehyde, ethyl acetoacetate, piperidine, glacial acetic acid, ethanol, and phenylhydrazine were purchased from Aldrich (Toluca, Mexico) and used as received. The ACS grade solvents were purchased from CTR (Jalisco, Mexico) and used without further purification. Melting points were measured on an Electrothermal Mel-Temp 1201D apparatus. IR spectra were collected using Varian 3100 FT-IR EXCALIBUR series spectrophotometer. ^1^H and ^13^C NMR spectra were recorded in a BRUKER Ultrashield plus 400 MHz in DMSO-d_6_ solutions using (CH_3_)_4_Si as an internal reference compound; chemical shifts (δ) are in ppm and coupling constants (*^n^J*_H-H_) in Hz. Mass analysis was performed by electro spray ionization in a high-resolution mass spectrometer Bruker micOTQF- QTOF instrument (Bruker Daltonik GmbH, Bremen, Germany). Absorbances were obtained with a Perkin Elmer Lambda 900 UV/Vis/IR spectrophotometer; emissions were obtained in an Edinburan F55 fluorometer, employing the stock solutions of 1 × 10^−3^ M of compounds **1a**–**d** and **2a**–**d**. The fluorescence spectra were recorded under 400 nm excitation (except in the compound **2d** was in 372 nm). The quantum efficiencies were obtained using an integration sphere of direct photon counting, relating the absorbed and emitted photons. The optical data in the solid state were obtained similarly.

### 2.2. General Methods of Synthesis and Characterization

3-Acetyl-2*H*-chromene-2-one (**1a**–**d**) were prepared following reported procedures [15,27] with modifications. 4-Substituted-salicylaldehydes, ethyl acetoacetate, and piperidine were dissolved in 15 mL of ethanol and placed under reflux at 78 °C for 24 h. After cooling, the solvent was removed by vacuum filtration, and the resulting solid was recrystallized from cold ethanol.

3-Acetyl-2-chromen-2-one (**1a**) [25]. Obtained from salicilaldehyde (0.42 mL, 4.1 mmol), ethyl acetoacetate (0.51 mL, 4.1 mmol), and piperidine (17.4 μL, 3 drops). White solid, yield = 0.62 g (80%), mp = 94–96 °C. λ Abs (nm) = 297 (THF), 296 (acetonitrile). IR (cm^−1^): ν (C-H) 2978, ν (C-H aromatic) 3047, ν (C=C) 1674–1450, ν (C=O) 1681, ρ (C-H f.p) 771. ^1^H-NMR (DMSO-d6), δ (ppm): 8.67 (1H, s, H-4), 7.95 (1H, d, ^3^J 8.3, H-5), 7.74 (1H, dd, ^3^J 7.7, 7.3, H-7), 7.48 (1H, d, ^3^J 7.7, H-8), 7.44 (1H, d, ^3^J 8.3, 7.3, H-6), 2.51 (3H, s, H-12). ^13^C-NMR (DMSO-d_6_), δ (ppm): 195.5 (C-11), 158.8 (C-2), 155.0 (C-9), 147.4 (C-4), 134.9 (C-7), 131.2 (C-5), 125.4 (C-6), 124.9 (C-3), 118.6 (C-10), 116.5 (C-8), 30.4 (C-12).

3-Acetyl-7-diethylamino-2-chromen-2-one (**1b**) [28,29]. Obtained from 4-diethylamino-salicylaldehyde (0.500 g, 2.6 mmol), ethyl acetoacetate (0.32 mL, 2.6 mmol) and piperidine (17.4 μL, 3 drops). Yellow solid, yield = 0.37 g (59%), mp = 156–158 °C. λ Abs (nm) = 426 (THF), 429 (acetonitrile), 424 (toluene), λ Em (nm) = 466 (THF), 477 (acetonitrile), 461 (toluene). IR (cm^−1^): ν (C-H) 2924, ν (C-H aromatic) 2962, ν (C=C) 1608–1411, ν (C=O) 1658, ν (C-N) 1188, ρ (C-H f.p) 759. ^1^H-NMR (DMSO-d6), δ (ppm): 8.4 (1H, s, H-4), 7.65 (1H, d, ^3^J 9.0, H-5), 6.78 (1H, dd, ^3^J 9.0, ^4^J 2.4, H6), 6.56 (1H, d, ^4^J 2.3, H-8), 3.9 (3H, s, H-12), 3.49 (4H, q, ^3^J 7.0, N (*CH_2_*CH_3_)_2_.), 1.14 (6H, t, ^3^J 7.0, N (CH_2_*CH_3_*)_2_.). ^13^C-NMR (DMSO-d6), δ (ppm): 194.6 (C-11), 160.3 (C-2), 158.7 (C-9), 153.4 (C-7), 148.0 (C-4), 132.8 (C-5), 115.5 (C-3), 110.6 (C-6), 107.9 (C-10), 96.3 (C-8), 44.8 (N (*CH_2_*CH_3_)_2_.), 12.8 N (CH_2_*CH_3_*)_2_.

3-Acetyl-7-hydroxy-2-chromen-2-one (**1c**) [30]. Obtained from 4-hydroxy-salicylaldehyde (0.500 g, 3.62 mmol), ethyl acetoacetate (0.42 mL, 3.6 mmol) and piperidine (17.4 μL, 3 drops). Green solid, yield = 0.68 g (89%), mp = 240–242 °C. λ Abs (nm) = 359 (THF), 356 (acetonitrile), λ Em (nm) = 460 (THF), 453 (acetonitrile). IR (cm^−1^): ν (C-H) 2927; ν (C-H aromatic) 3032, ν (C=C) 1678–1450, ν (C=O) 1678; ν (O-H) 3194, ρ (C-H f.p) 763. ^1^H-NMR (DMSO-d6), δ (ppm): 8.57 (1H, s, H-4), 7.65 (1H, d, ^3^J 8.6, H-5), 6.86 (1H, dd, ^3^J 8.6, ^4^J 2.2, H-6), 6.73 (1H, d, ^4^J 2.2, H-8), 2.63 (3H, s, H-12). ^13^C-NMR (DMSO-d6), δ (ppm): 195.1 (C-11), 165.1 (C-2), 159.5 (C-9), 157.8 (C-7), 148.2 (C-4), 133.1 (C-5), 119.4 (C-3), 114.8 (C-6), 111.1 (C-10), 102.2 (C-8), 30.4 (C-12).

3-Acetyl-7-methoxy -2-chromen-2-one (**1d**) [31]. Obtained from 4-methoxy-salicylaldehyde (0.500 g, 3.28 mmol), ethyl acetoacetate (0.41 mL, 3.3 mmol) and piperidine (17.4 μL, 3 drops). Yellow solid, yield = 0.63 g (87%), mp = 172–174 °C. λ Abs (nm) = 356 (THF), 354 (acetonitrile). IR (cm^−1^): ν (C-H) 2978, ν (C-H aromatic) 3032, ν (C=C) 1666–1462, ν (C=O) 1670, ν (C-O-C) 1126, ρ (C-H f.p) 767. ^1^H-NMR (DMSO-d6), δ (ppm): 8.62 (1H, s, H-4); 7.86 (1H, d, ^3^J 8.6, H-5), 7.02 (1H, dd, ^3^J 8.6, H-6), 7.04 (1H, s, H-8), 2.56 (3H, s, H-12), 3.9 (3H, s, OMe). ^13^C-NMR (DMSO-d6), δ (ppm): 195.1 (C-11), 165.3 (C-2), 159.3 (C-9), 157.5 (C-7), 148.0 (C-4), 132.6 (C-5), 120.8 (C-3), 113.9 (C-6), 112.2 (C-10), 100.7 (C-8), 56.7 (OMe), 30.4 (C-12).

Compounds **2a**,**c** were prepared according to previous reports [31]. Compounds **2b**,**d** have not been reported yet. The corresponding 4-substituted compound **1a**–**d**, phenylhydrazine, and glacial acetic acid were added to 10 mL of ethanol and refluxed at 78 °C for 4 h. The solvent was removed by gravity filtration, and the solid was washed with ethanol, recrystallized from ethanol at room temperature, and then characterized.

3-(1-(2-Phenylhydrazone) ethyl) 2-chromen-2-one (**2a**) [32]. Obtained from **1a** (0.300 g, 1.07 mmol), phenylhydrazine (0.105 mL, 1.07 mmol) and glacial acetic acid (13.6 μL, 3 drops). Orange solid, yield = 0.39 g (74%), mp = 184–186 °C. λ Abs (nm) = 277 (THF), 275 (acetonitrile). IR (cm^−1^): ν (C-H) 2978, ν (C-H aromatic) 3032, ν (C=C) 1597–1489, ν (NH) 3294, ν C=N (1597), ρ (C-H f.p) 732. ^1^H-NMR (DMSO-d6), δ (ppm): 9.46 (NH, s), 8.20 (1H, s, H-4), 7.84 (1H, d, ^3^J 7.8, H-5), 7.60) (1H, dd, ^3^J 7.9, 7.4, H-7), 7.41 (1H, d, ^3^J 8.0, H-8), 7.36 (1H, dd, ^3^J 7.9, 7.5, H-6), 7.21–7.27 (4H, *o, m*, -Ph), 6.79 (1H, *p*, ^3^J 7.0, -Ph). 2.23 (3H, s, H-12), ^13^C-NMR (DMSO-d6), δ (ppm): 160.4 (C-2), 153.4 (C-9), 146.0 (C-11), 139.6 (C-4), 139.1 (*ipso*, -Ph), 132.0 (C-7), 129.3 (C-5), 129.2 (*m*, -Ph), 127.8 (C-3), 125.0 (C-6), 119.8 (C-10), 119.7 (p, -Ph), 116.3 (C-8), 113.5 (*o*, -Ph), 15.5 (C-12).

7-(Diethylamino)-3-(1-(2-phenylhydrazone)ethyl)2-chromen-2-one (**2b**). Obtained from **1b** (0.300 g, 1.15 mmol), phenylhydrazine (0.113 mL, 1.15 mmol) and glacial acetic acid (13.6 μL, 3 drops). Orange solid, yield = 0.32 g (65%), mp = 236–238 °C. λ Abs (nm)= 411 (THF), 413 (acetonitrile), 408 (toluene), λ Em (nm) = 479 (THF), 494 (acetonitrile), 469 (toluene). IR (cm^−1^): ν (C-H) 2900; ν (C-H aromatic) 2966; ν (C=C) 1589–1408, ν (NH) 3302, ν C=N (1589), ρ (C-H f.p) 748. ^1^H-NMR (DMSO-d6), δ (ppm): 9.23 (NH, s), 8.01 (1H, s, H-4), 7.56 (1H, d, ^3^J 8.9, H-5), 7.22-7-20 (4H, *o, m*, -Ph), 6.73 (1H, *p*, -Ph), 6.71 (1H, d, ^3^J 8.9, H-6), 6.55 (1H, s, H-8), 2.20 (3H, s, H-12), 3.45 (4H, q, ^3^J 6.9, (N *(CH_2_CH_3_*)_2_), 1.14 (6H, t, ^3^J 7.0, N (*CH_2_CH_3_*)_2_), ^13^C-NMR (DMSO-d6), δ (ppm): 160.9 (C-2), 156.4 (C-9), 150.9 (C-7), 146.4 (C-11), 140.4 (*ipso*, -Ph), 140.6 (C-4), 130.2 (C-5), 129.2 (*m*, -Ph), 120.1 (C-3), 119.3 (*p,* -Ph), 113.4 (m, -Ph), 109.7 (C-6), 108.6 (C-10), 96.6 (C-8), 44.5 (N *(CH_2_CH_3_*)_2_), 15.7 (C-12), 12.8 (N (*CH_2_CH_3_*)_2_). MS-MS (ESI) *m*/*z* = 350.1866 [M + H]^+^ (experimental), 350.1869 (calculated).

7-(Hydroxy)-3-(1-(2-phenylhydrazone)ethyl)-2-chromen-2-one (**2c**) [26]. Obtained from **1c** (0.300 g, 1.46 mmol), phenylhydrazine (0.144 mL, 1.46 mmol) and glacial acetic acid (13.6 μL, 3 drops). Yellow solid, yield = 0.36 g (65%), mp = 228–230 °C. λ Abs (nm) = 278 (THF), 276 (acetonitrile), λ Em (nm) = 465 (THF). IR (cm^−1^): ν (C-H aromatic) 3032; ν (C=C) 1593–1442, ν (NH) 3325, ν (O-H) 3221 ν (C=N) 1597, ρ (C-H f.p) 748. ^1^H-NMR (DMSO-d6), δ (ppm): 10.62 (1H, s, OH) 9.31 (NH, s), 8.09 (1H, s, H-4), 7.66 (1H, d, ^3^J 8.4, H-5), 6.81 (1H, d, ^3^J 8.4, H-6), 6.74 (1H, s, H-8), 7.22 (4H, *o, p*, -Ph), 6.77 (1H, *p*, -Ph), 2.20 (3H, s, H-12). ^13^C-NMR (DMSO-d6), δ (ppm): 161.6 (C-7), 160.4 (C-2), 155.5 (C-9), 146.0 (C-11), 140.4 (C-4), 139.7 (*ipso*, -Ph), 130.6 (C-5), 129.2 (*m*, -Ph), 123.3 (C-3), 119.6 (*p*, -Ph), 113.9 (C-6), 113.4 (*o,* -Ph), 112.2 (C-10), 102.2 (C-8), 15.6 (C-12).

7-(Methoxy)-3-(1-(2-phenylhydrazone)ethyl)-2-chromen-2-one (**2d**). Obtained from **1d** (0.300 g, 1.48 mmol), phenylhydrazine (0.148 mL, 1.46 mmol) and glacial acetic acid (13.6 μL, 3 drops). Yellow solid, yield = 0.41 g (65%), mp = 228–230 °C. λ Abs (nm) = 278 (THF), 275 (acetonitrile), λ Em (nm) = 467 (THF). IR (cm^−1^): ν (C-H) 2997, ν (C-H aromatic) 3020, ν (C=C) 1612–1496, ν (NH) 3309, ν (C=N) 1608, ρ (C-H f.p) 736. ^1^H-NMR (DMSO-d6), δ (ppm): 9.36 (NH, s), 8.15 (1H, s, H-4), 7.77 (1H, d, ^3^J 8.65 H-5), 7.23–7.22 (4H, *o, m*, -Ph); 7.0 (1H, s, H-8), 6.98 (1H, d, ^3^J 8.6, H-6), 6.79 (1H, *p,* -Ph), 3.88 (3H, s, OMe), 2.21 (3H, s, H-12), ^13^C-NMR (DMSO-d6), δ (ppm): 162.8 (C-2), 160.3 (C-2), 155.4 (C-9), 146.1 (C-11), 140.1 (C-4), 139.5 (*ipso*, -Ph), 130.3 (C-5), 129.3 (*m*, -Ph), 124.3 (C-3), 119.6 (*p*,-Ph), 113.5 (C-6), 113.3 (*o*, -Ph), 113.3 (C-6), 100.7 (C-8), 56.4 (OMe), 15.5 (C-12). MS-MS (ESI) *m*/*z* = 309.1235 [M + H]^+^ (experimental), 309.1239 (calculated).

### 2.3. Determination of Quantum Yield

Fluorescence quantum yield (*Φ*) for coumarins (**1a**–**d**) and hydrazones (**2a**–**d**) were measured through the integration sphere direct excitation method, with “Direct Excitation” measurements, which record the scatter and the emission of the sample being directly excited by the radiation from the excitation monochromator only. The absolute fluorescence quantum yield (*Φ*) is the ratio of the number of photons emitted to the number of photons absorbed, according to Equation (1).
(1)$$\phi =\frac{N^{em}}{N^{abs}}$$

The absolute fluorescence quantum yield (*Φ_Dexc_*) was calculated with the direct excitation method according to Equation (2).
(2)$$\phi_{Dexc}=\frac{E_{B}-E_{A}}{S_{A}-S_{B}}$$
where *E* denotes the emission region of sample and solvent, *S* the excitation scatter region of sample and solvent, and *A*, *B* the experimental setup and integral of the scans of samples and solvent.

For calculating the integrals, the selection of the integral regions, and the final calculation of *Φ_Dexc_*, we used the quantum yield wizard supplied with the equipment software [33].

### 2.4. Computational Details

Geometry optimizations were carried out using the framework of density functional theory implemented in ORCA software [34]. For the exchange and correlation, the PBE0 hybrid functional [35,36] was combined with a complete electronic basis set aug-cc-pVDZ [37]. A vibrational frequency analysis was effectuated to verify minima energy states; only positive values were found in all cases. The HOMO-LUMO gap energies were calculated, and frontier orbitals were plotted using Chemcraft [38]. Excited states were determined using TD-DFT [39,40,41] under the same level of theory PBE0/aug-cc-pVDZ [42], and solvent effects were included using a conductor-like polarizable model (CPCM) [43,44] considering THF, acetonitrile, and toluene.

## 3. Results and Discussion

### 3.1. Synthesis and Structural Characterization

Coumarins **1a**–**d** were synthesized by the Knoevenagel [28] condensation of ethyl acetoacetate with the corresponding 4-substituted salicylaldehyde (Figure 1), with yields higher than 80%. Afterward, the functionalization of 3-acetyl-coumarin derivatives **1a**–**d** with phenylhydrazine was performed through acid catalysis with glacial acetic acid to obtain compounds **2a**–**d** (Figure 1). The experimental yields are in the 65–74% range, according to precise reports about the synthesis of the same compounds [2,29,32,45].

The H-4 signal in the ^1^H NMR spectra indicated the formation of compounds **1a**–**d** (Appendix A), appearing in the 8.40–8.67 ppm range [31]. The functionalization with phenylhydrazine in compounds **2a**–**d**, generated an anisotropic effect perpendicularly extended from the phenyl group of the hydrazone fragment to the pyran ring, causing the protection of H-4 and shifting the signal to lower frequencies in the 8.00–8.15 ppm range. The NH proton in the phenylhydrazone was shifted to high frequencies appearing in the 9.31–9.46 ppm range; whereas the ^13^C-NMR signal was shielded from 194.6–195.5 ppm, for compounds **1a**–**d**, to the range of 146.0–146.4 ppm in compounds **2a**–**d** (Appendix A). The IR spectra of compounds **1a**–**d** showed the C=O band in the 1658–1681 cm^−1^ range. Meanwhile, compounds **2a**–**d** showed the characteristic hydrazone C=N and N-H bands in the 1589–1608 cm^−1^ and 3294–3325 cm^−1^ ranges, respectively. Selected ^1^H and ^13^C NMR values and IR wavenumbers are listed in Table 1.

### 3.2. UV-vis Absorption and Photoluminescence Spectra

The UV-vis and fluorescence spectroscopic data of compounds **1a**–**d** and **2a**–**d**, measured in tetrahydrofuran and acetonitrile solvents at room temperature, are listed in Table 2. Their scarce solubility in solvents, such as ethanol, DMF, and water, limited the tests in these solvents. The absorption and emission spectra in solution are shown in Figure 1 and Figure 2, respectively. The maximum absorbance bands are attributed to π → π* and n → π* electronic transitions. The absorption band in **1b**–**d** showed a red-shift due to the insertion of electron-donor substituents in the C-7 position. The largest red-shift in **1b** is attributed to the strong electron-donor character of the diethylamino group [4,9]. Similar behavior is conserved in phenylhydrazone derivatives, wherein **2b** shows the largest bathochromic effect in the absorption band. The molar absorptivity depends on the solvent used, obtaining higher values with acetonitrile than with THF. In contrast to the full palette observed in the absorption, compounds **1a**,**d** and **2a** showed no emission band. The maximum wavelength of the coumarin emission band is around 453–494 nm. The increase of emission is related to the nature of the substituent in C-7 following the order: N (CH_2_CH_3_)_2_ > OH > OCH_3_ > H, in agreement with the strength of the substituent electron-donor properties [4,9]. Compound **1b** showed the largest quantum yield (*Φ*_F_) with values above 80%.

On the other hand, a brief comparison between the photophysical properties of coumarins **1a**–**d** with the corresponding values of hydrazones **2a**–**d** allowed us to conclude that coumarin functionalization in the C-3 position with hydrazine caused fluorescence quenching with null fluorescence yields. This result is explained by the diminished electron acceptor capabilities of the resulting hydrazone functionality.

Compounds **1b**,**d**, and **2c** showed the best photophysical properties in the solid-state. Their fluorescence spectra are shown in Figure 3, and their photophysical properties are listed in Table 3. The emission λ_max_ of coumarin **1b** was red-shifted to 595 nm in comparison with the solution (477 nm), and unexpectedly, compound **1d** turned emissive, showing a band at 540 nm, although with low *Φ*_F_ values (<6.6%) in both cases. In contrast, hydrazone 2c has emissive properties compared to the solution: emission at 582 nm and *Φ**_F_* value of 52.6. These results revealed that the intermolecular interactions in the solid-state favor the emissive relaxation mechanism from an excited state. The X-ray molecular structures of compounds **1a**,**b**,**d** are known [46,47,48,49]. Hence, a brief analysis of their photoluminescent properties in relation with their molecular structure in the solid state seems appropriate. The powder X-ray diffractogram of **1a**,**b**,**d** was obtained and compared to the single crystal reported ones, shown in Appendix A. In the solid, the free rotation of the pendant –OMe and 3-Ac groups is restricted due to intermolecular hydrogen bonds, leading to a coplanar system and increasing the emissive properties of **1d**. Therefore, the solid-state exerts an aggregation-induced emission (AIE) effect [46] on **1d**. In the solid, the molecules of **1d** are π-stacked along the (100) direction, at a distance of 3.509 Å between the benzo fused (Bz) and lactone (L) rings centroids (Cg (Bz)-Cg (L)) [47], in agreement with medium strength π-D-π-A interactions arranged in the head-to-tail fashion [50]. It is worth mentioning that the Bz ring acts as the donor and the L ring as the acceptor. In contrast, coumarin **1a** crystallizes in two forms, triclinic (form A) and monoclinic (form B). Form A (herein obtained) is a head-to-head stacked, while form B generates head-to-tail π-stacked dimers with long Cg (Bz)-Cg (L) distances of 4.77 Å and 3.98 Å, respectively [48]. On the contrary, an aggregation-caused quenching (ACQ) effect is observed on **1b**. In the crystal lattice, the molecules of **1b** are arranged in dimers through antiparallel type carbonyl–carbonyl interactions [51] at Cg (Bz)-Cg (L) distance of 6.38 Å and the amino-ethyl groups pointing to opposite directions [49]. The free rotation of the pendant –NEt_2_ and 3-Ac groups in **1b** is restricted as in **1d**, but with the contrary effect. This brief analysis supports that the *Φ*_F_ in the solid-state depends on the crystal lattice arrangement. Coumarin molecules that show head-to-tail π-staking arrangement with a Cg(Bz)-Cg(L) distance shorter than 3.5 Å seem to favor large *Φ*_F_ values. The large value of *Φ*_F_ of hydrazone **2c** in the solid-state is also explained as an AIE effect. Intermolecular hydrogen bonding, probably between the –OH and the C=N functionality, should favor an entire head-to-tail π-staking arrangement in the crystal lattice with the appropriate Cg (Bz)-Cg (L) distance, as mentioned earlier.

Compounds **1**–**5** have been used as fluorescent dyes for cellular imaging (**1**), indicators of bathochromic effects with red emissions (**2**), chemosensors for anions (**3**), for the study of nonlinear optical properties (**4**), and observation of highly fluorescent compounds depending on the substituents (**5**). Their photophysical properties are listed in Table 4. 7-Diethylamino substituted coumarin **1b** showed better photophysical properties (*Φ*_F_) in solution than compounds **1**–**5**, previously reported and shown in Figure 4 [6,7,10,16,52]. Then, compound **1b** in solution and **2c** in the solid-state could be evaluated in similar applications, since they have even better optical properties, and their synthesis is simpler and with higher chemical yields than the reported compounds **1**–**7**.

### 3.3. Theoretical Calculations

The theoretical data of the molecular orbitals allowed us to explain the spectroscopic behavior and the electronic effects in the molecules. The HOMO (H), LUMO (L), H-L gap, and Eg^op^ energy values are listed in Table 5. The H-L gap energies are similar among them. Nevertheless, the smallest value corresponds to compound **1b**, whose λ_max_ is displaced towards the red. Likewise, the H-L gap represents the excited state electronic transition [6,38] from S_0_ to S_1_. Compounds **2a**–**d** show low H-L gap values similar to coumarins **1a**–**d**. The theoretical values, listed in Table 5, agree with the experimental optical values.

The optical gap values (Eg^op^) for coumarins rank from 2.90 to 4.18 eV, and they are smaller than the theoretical data (3.84–4.47 eV); where compound **1b** shows the smallest Eg^op^ value, indicating good ICT capability. Eg^op^ values of compounds **2a**–**d** show low energy values (3.03–4.52 eV) like coumarins. These results indicate that the absorbance and emission red-shifts are due to the good electronic efficiency in their D-A structure. The theoretical and experimental H-L gap values are in agreement with the experimental optical properties and increase their possibility to function as semiconductor compounds [55]. The experimental values are comparable with the data for well-known luminophores **3**, **6**, and **7** listed in Table 6 and whose structures are shown in Figure 5.

The data of compounds **1b**–**d** showed HOMO-LUMO interactions (Appendix A) in a higher percentage of contribution energy due to the ED groups generating n → π* transitions, in contrast to compound **1a**, which causes HOMO-1-LUMO π → π* transitions, in agreement with the experimental data.

Compounds **2a**–**d** showed π → π* and n → π*transitions represented by the HOMO-LUMO, HOMO+1-LUMO-1 orbitals (Appendix A), indicating a D-A behavior of the molecules which is consistent with the experimental data.

Figure 5 shows the FMOs (frontier molecular orbitals) of compounds **1a**–**d**. The structures show a planar geometry through the benzopyran ring. The HOMO-LUMO and HOMO-1-LUMO electronic distributions indicate the presence of π → π* and n → π* transitions, confirming the excitation of the FMOs from S_0_ to S_1_.

In compounds **2a**–**d**, the phenylhydrazine group acts as an ED group. In this case, it represents the energy of the HOMO, and often, the whole part of the coumarin where the substituent in the benzopyrone ring acts as EA group, represented by the energy of the LUMO (Figure 6). The previous result agrees with the experimental findings, suggesting that the electronic cloud of the ED moiety moves towards the EA, which is from the phenylhydrazine to the coumarin. The hydrazo-coumarins **2a**–**d** showed low quantum yields and unobservable emissions because the electronic cloud does not include the ED group which is interacting with the solvents.

Compound **2c** shows different behavior in the FMOs, and its electron density is both as HOMO and LUMO in the hydrazone group, generating only n → π* type transitions, which agrees with its behavior in the solid-state, since the solvent generates the quenching of the fluorescence in solution.

## 4. Conclusions

Molecules **1a**–**d** and **2a**–**d** can be good candidates for semiconductor compounds suitable to be used in several optical applications. The photophysical properties of coumarins **1a**–**d** and their hydrazone derivatives **2a**–**d** were compared. The bandgap indicates the efficient ICT, which confirms the D-A type structure. Compounds **1b** and **2c** presented the best optical properties. Compound **1b** showed absorbances at λ > 420 nm and emission at λ > 460 nm in solution and quantum yields >80%. Compound **2c** also showed good emission values in the solid-state due to the push-pull system, supported by the π-π interactions. Compound **1d** turned emissive in the solid-state due to the π-π- interactions in the crystal network. This work also confirms that small coumarins derivatives possess photoluminescent properties which can be tuned in both solution and in the solid-state.

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
