# Peer review of "Synthesis, Optical Characterization in Solution and Solid-State, and DFT Calculations of 3-Acetyl and 3-(1′-(2′-Phenylhydrazono)ethyl)-coumarin-(7)-substituted Derivatives"

_molecules, 2022, doi:10.3390/molecules27123677_

Round 1
Reviewer 1 Report
The article describes fluorescence characterization of coumarine derivatives. I recommend to accept the article after the minor revision.
The following aspects should be clarified and corrected:
line 25: " structural arrangement" - in the article the solid state structure is not discussed, therefore authors should avoid "structural"; the same in line 55 & 197
line 46: [ref], please include references
line 175: excited not exited
line 220: chapter 3.2 please explain why the fluorescence absorbance is not present for some toluene solvents. This might be wider discussed.
line 234 : electron-donor properties? does it mean electron-donating properties?
line 238: electron donating - group substitued in C7 position
table 2 layout of table 2 should be corrected
line 306: the caption of Figure 5 should be corrected because references for structures should be included here
line 345: the abbreviation FMO should be explained
Reviewer 2 Report
In this manuscript, Martinez and co-workers are reporting a series of coumarin and hydrazone-based fluorescence probes while studying their optical properties in both solid state as well as in solvent. In addition, authors further report DFT calculations to explain properties of the molecules. The manuscript content is interesting. However, the current version of the manuscript has significant issues with the scientific writing and the soundness. There are major flow issues with the current writing. Authors should pay significant attention to their writing while explaining experimental findings. I strongly advise authors to use editorial service if necessary. In addition, I would also recommend following modifications to the current version of the manuscript.
(1). All figure captions of the manuscript as well as in the supporting information should be modified. A figure caption should accurately describe the data represented. Some of the figure captions does not make sense. Especially while representing optical spectra, specify the concentrations, excitation wavelength and the temperature.
(2). In the first page of the manuscript lines 35 to 35 is extremely confusing and the sentence lacking a meaning. Authors should carefully revise this paragraph.
(3). Authors should provide at least several examples for the ICT fluorophore systems in the introduction section and also should provide a schematic representation to explain how ICT occurs.
(4). The abstract of the manuscript should revise carefully, as the current abstract has multiple grammatical and flow errors.
(5). In the experimental section "A.C.S. grade" must be corrected as "ACS grade"
(6). Did authors really used 1 mM concentration for spectroscopic studies? This concentration is extremely high. The standard concentration for spectroscopic study is 10 μM. The current working concentration is extremely high. If authors really use such a high concentration, this study is highly questionable, and all spectroscopic data should be repeated at 10 μM concentration.
(7). Coupling constants mut be shown for proton NMR spectra.
(8). Solvent optical properties must be also obtained in EtOH or MeOH, DMF or DMF and in water. This information is vital understand behavior/environmental sensitivity of the probes.
(9). In the table 2, column with "solvent" has textual errors. In addition, this table must be completed, and it is confusing why some data missing from the table.
(10). Figure 2 and 2 representation quality is very poor. Figure one must represent the absorbance is the y axis instead of the molar absorptivity. Figure 2 has different y axis. It is clearly indicating the manuscript has been prepared in a hurry and authors have not paid enough attention to the data representation. What are the working concentrations here?
(11). In page 8 authors explains aggregation of the probes but no experimental data has been provided.
(12). What is the application of these probes having a large fluorescence quantum yield in the solid state compared to the solution?
(13). Figure 9, lines 291 to 300 does not imply a clear meaning. This text should be removed or modified accordingly.
(14). In table 6, HOMO-LUMO energy calculations are provided for the probes. Can authors calculate the theoretical excitation wavelength in nm based on this data and compare with the experimental data?
(15). Figures 6 and 7 should combined concisely.
(16). The conclusion is very unclear and disorganized. This must be revised thoroughly.
Round 2
Reviewer 2 Report
Recommending to accept in the current form.